# Development and Validation of the Pain and State of Health Inventory (PHI): Application for the Perioperative Setting

**DOI:** 10.3390/jcm10091965

**Published:** 2021-05-03

**Authors:** Julia Stuhlreyer, Regine Klinger

**Affiliations:** Center for Anesthesiology and Intensive Care Medicine, University Medical Center Hamburg-Eppendorf, 20251 Hamburg, Germany; r.klinger@uke.de

**Keywords:** PHI, SBI, pain, perioperative setting, questionnaire, inventory, acute pain, postoperative pain, quality of recovery, pain treatment

## Abstract

Currently, general measurements and evaluations of the quality of recovery are difficult because no adequate measuring tools are available. Therefore, there is an urgent need for a universal tool that assesses patient-relevant criteria—postoperative pain, state of health, and somatic parameters. For this purpose, a pain and state of health inventory (PHI, Schmerz- und Befindlichkeitsinventar (SBI) in German) has been developed. In this study, we describe its development and validation. The development phase was led by an expert panel and was divided into three subphases: determining the conceptual structure, testing the first editions, and adjusting the inventory for a finalized edition. For the purpose of validation, the PHI was filled in by 132 patients who have undergone total knee replacement and was analyzed using principal component analysis. Construct validity was tested by correlating the items with validated questionnaires. The results showed that the inventory can test pain, state of health, and somatic parameters with great construct validity. Furthermore, the inventory is accepted by patients, map changes, and supports to initiate adequate treatment. In conclusion, the PHI is a universal tool that can be used to assess the quality of recovery in the perioperative setting and allow immediate intervention.

## 1. Introduction

Evaluating the quality of recovery in the perioperative setting is considered very important to ensure the best possible treatment for patients. Therefore, many studies have focused on the quality of recovery [1,2]. However, the criteria for such quality are not overall the same because they can be defined and assessed in various ways [3,4]. Depending on the underlying criteria, different questions related to recovery can be answered. However, even with the same criteria, the answers usually differ depending on the response options and respondents. In addition, opinions of the persons involved in the surgery and recovery process might also vary regarding the outcome quality of the same treatment [5]. Traditional measures for assessing treatment outcomes usually include morbidity and mortality [6]. In addition, common tools which are used to assess the quality of recovery (e.g., QoR-15 [7]) focus on the frequency of conditions (e.g., feeling of general well-being) but do not explore the intensity. Newer measures should go beyond frequency measures and should include the intensity of outcomes that are especially important to the patients, and should provide the possibility to also include the perspective of the treatment team [2,6].

Generally, evaluating the quality of recovery is important not only because it has a direct impact on the immediate treatment quality, but also because it prevents negative long-term effects, such as chronic pain or morbidity. Postoperative pain is a relevant marker of postoperative healing, and severe postoperative pain is a risk factor for the development of chronic pain [8,9]. It should be noted that the quality of recovery is associated with economic consequences, which are often assessed using cost–benefit analyses [10]. Therefore, all involved stakeholders should be interested in a high quality of recovery. To prevent the persistence of postoperative pain and assess the treatment quality and improve it, it is necessary to develop, both internationally and nationwide, guidelines for evaluating the quality of recovery. Such a need and demand for guidelines for quality assurance and effectiveness measurement is partly based on the approach of evidence-based medicine (EBM) [11]. Generally, EBM emphasizes the relevance of improving medical treatment based on scientific evidence and emphasizes shared decision-making. Patients should be able to communicate their needs and judgments, which should be further included in outcome measures pertaining to the quality of recovery. Hence, the criteria for the quality of recovery should include outcome measures relevant for the patient and should be representative of the whole perioperative setting [12]. Thus, quality measurement can be viewed as quality of care in the perioperative setting and should include and be centered around the patients’ needs [13,14].

The relevant outcomes for patients are pain [2,15], state of health [16], and somatic parameters [5,15]. Although pain is considered one of the most severe problems in surgical practice and 30–80% [17,18,19] of patients experience severe postoperative pain, postoperative pain diagnosis and pain therapy are rarely performed [20,21]. Early and effective acute pain treatment leads to a better general state of health and faster mobilization and, therefore, decreased vulnerability to thrombosis and pneumonia [22]. Hence, early acute pain treatment is a requirement for a fast recovery and reduces the postoperative risk for morbidity and mortality [23]. Generally, acute pain is a risk factor for chronic pain, which is considered a serious global wide-ranging interdisciplinary problem and can only be ameliorated if pain experiences and emotional states are paid more attention [24]. The relationship between pain experiences and emotional states, such as depression and anxiety, is particularly evident in chronic pain [25,26]. However, the emotional state is also relevant in acute pain [27], especially in acute postoperative pain [28,29,30,31,32]. On the one hand, pain and depression are considered risk factors for acute pain because depression and anxiety lead to an increased perception of pain. On the other hand, a prolonged duration of acute pain can lead to increased mood dysregulation [27]. Thus, the relationship between the state of health, including the emotional state, and pain experiences goes beyond simple cause and effect [33].

One reason for the inadequate postoperative pain treatment is that no suitable tool is currently available to perform an appropriate pain diagnosis and assess the state of health within the perioperative setting [4]. Although numerous tools are available for assessing either depression, anxiety, or pain, none of them combines relevant features and takes the specific perioperative setting into account. If relevant outcomes are to be assessed within the perioperative setting, then so far it is necessary to apply a comprehensive battery of questionnaires focusing on the emotional state and additional questionnaires focusing on pain. However, the use of such a large number of questionnaires usually leads to displeasure for the patients, which can limit the informative value of the provided answers. Therefore, to determine whether the quality of recovery is sufficient, it is important to have a universal tool that can help compare and improve the quality of recovery for surgical procedures. For this purpose, a pain and state of health inventory (PHI, also called Schmerz- und Befindlichkeitsinventar (SBI) in German) was developed by Regine Klinger and Florian Krug [34]. This tool was designed as a universal tool for clinical standard diagnoses in the perioperative setting and includes guidelines from EBM. Using this PHI allows the assessment of surgery-relevant outcomes in the perioperative setting with only one measurement tool.

The aim of this study is to demonstrate the development of the PHI and validate it as a general tool designed for the perioperative setting. In this context, in this study, we investigated whether the PHI is widely accepted, easy to understand, sensitive to changes within the perioperative setting, and of general use in everyday clinical practice.

## 2. Materials and Methods

In this study, we focus on the development and validation of the PHI. The first part of this article describes the development of the PHI, and the second part describes the validation of its finalized edition.

### 2.1. Development Process

The development process of the PHI consisted of three main steps: (i) determining the conceptual structure formulating corresponding items, (ii) testing the first editions, and (iii) adjusting the inventory for a finalized edition. Both the conceptual structure and items were designed by experts in the perioperative setting and were validated with external criteria. The consulted experts were specialists in orthopedics and trauma surgery, and specialists in psychological pain psychotherapy, both with several years of experience in operative and postoperative management. The strategy to base the items on expert rating was chosen to facilitate a discussion about the included items and to adjust the inventory based on patients’ suggestions and recommendations. The items are arranged into different subgroups: pain, state of health (including emotional states), and somatic parameters. To cover the whole perioperative setting, the PHI consists of an entry version, which is filled out preoperatively, and a progress version, which is filled out postoperatively. Hence, the version of the inventory indicates the occasion the patient receives the inventory. However, all patients always receive both versions.

#### 2.1.1. Conceptual Structure

The conceptual structure consists of different subsections: pain, state of health (including emotional states), and somatic parameters.

#### 2.1.2. Pre- and Postoperative Pain

For pre- and postoperative pain, the specification of pain perception, including intensity, location, duration, and frequency, is considered to be particularly relevant [35]. For this purpose, items covering corresponding specifications were formulated with an expert panel. The applied research tools were selected based on commonly applied tools in practice, so that they are simple and fast for the patients to fill out and easy for health care practitioners to interpret. The most prominent validated tools to assess pain intensity are the Numeric Rating Scale (NRS), the Visual Analogue Scale (VAS), the Verbal Rating Scale (VRS), and the Faces Pain Rating Scale (FPS) [36,37]. However, usually, the NRS have more validity than the other pain intensity measure tools [36,38]. Hence, to assess the intensity of pain, the Numeric Rating Scale (NRS) was applied [36,39,40]. Pain frequency was assessed by applying answer options that are usually applied in clinical practice [41]. To assess the location of pain, the most prominent tools are pain drawings or a default list of painful areas. The downside of pain drawings is that the interpretation can be challenging. If not given exact specification, patients complete pain drawings differently [42,43]. To facilitate the interpretation, an unambiguous default category system of painful areas was developed [44]. To assess pain duration, the patients were asked to report the number of days, months and/or years they have experienced pain [45].

#### 2.1.3. State of Health

To assess the state of health (including emotional states), relevant items were formulated with the expert panel by considering commonly used tools. Generally, most questionnaires pertaining to the postoperative emotional state deal with anxiety and depression [46]. Measurement tools assessing multidimensional emotional states are usually designed to detect psychopathological cases. On the basis of a mood-related questionnaire, mood adjectives were formulated with answer options scored on an 11-point NRS.

#### 2.1.4. Somatic Parameters

Somatic parameters are an important aspect for postoperative recovery for different surgeries [47,48,49,50]. Hence, diet [51,52] and mobility [23] are considered to be particularly important for recovery and have, therefore, been added to the item pool. The items should describe the extent to which a patient has exercised and the status of their nutrition. The items included were twofold: (i) patients were asked about the diet they received on a particular day and their desired diet, and (ii) patients were asked about their current level of exercise and their desired level of exercise.

#### 2.1.5. Application of the Inventory

To capture changes in the perioperative setting, it was planned to design an inventory that would be used preoperatively to record the patient’s preoperative state, and also postoperatively to facilitate patient monitoring. The aim is to enable the application of the inventory based on requirements and needs of the hospital ward, so that the inventory could be applied daily postoperatively or on certain days postoperatively.

#### 2.1.6. Preliminary Studies to Test the First Editions

To validate the corresponding items, the PHI was supplemented with the Pain Perception Scale, Hospital Anxiety and Depression Scale, German Version [53], Sensitivity Scale [54], List of Adjectives [55], and Profile of Mood States [56]. The first edition of the inventory was tested on 28 patients (17 female, 11 male) who have undergone surgery at the German University Medical Center Lübeck. It was observed that the length of the questionnaire battery was not very well accepted, in addition to great variations in responsiveness between the items. The questionnaire was handed out with questions regarding changes and concerns by the patients. To encourage patients to raise concerns and comments, patients were asked in a personal interview postoperatively about the individual items and the items they would have liked to be additionally included. The questionnaires applied for validation were rated by the patients as too long, difficult to understand, and suggestive. However, the first test of the PHI revealed that the items from the PHI itself were mostly widely accepted, and patients who have undergone surgery confirmed that the items covered the areas that they regarded as important for their recovery process. The PHI was adjusted based on the raised concerns and recommendation of the patients; due to the reason that length was a primary concern, the list of emotional states in the form of adjectives was shortened from sixteen to ten adjectives in total. Furthermore, the rating system was equal to the Profile of Mood States 7-scale Likert scale. This scale was changed to an NRS 0–10 to have an equal rating scheme throughout the inventory to simplify answering the different items. Additionally, an assessment tool for external assessment (physicians, nurses, and physiotherapists) was implemented to compare the patients’ perspective with external observations and help rate the perioperative progress.

The second edition was again tested at the German University Medical Center Lübeck on 111 patients (47 female, 64 male) who have undergone surgery. Special attention was again paid to truly include the patients’ perspectives. Therefore, patients were encouraged to suggest relevant items and raise concerns or write comments. As in the first edition, patients received additional questions to raise concerns and were asked in a personal interview about the individual items and if they missed relevant items. The results showed that the patients were content with the included subsections and items. They confirmed that their recovery criteria were covered in the PHI. On the basis of the comments made by the patients and the results obtained from the external observation, the PHI was adjusted and finalized, followed by a further validation step (see Section 2.2). One greater change was that, based on expert opinion, a new subscale called “general condition” was added to the inventory, whose scale was derived from Clinical Global Impression [57]. This scale enables comparing between the self-perception of the state of health and the perception of others. Furthermore, the adjectives were further reduced to seven adjectives, which are asked pre- and postoperatively in the next edition of the inventory. The questions for mobility and diet were not very well understood, which led to a change in the wording of the items.

#### 2.1.7. Finalized PHI

Based on the results from the second preliminary study, a finalized version was created. The items of the finalized inventory were structured as follows (Figure 1). The inventory was divided into an entry and a progress version. The entry version was handed out preoperatively, and the progress version was handed out postoperatively. Items were grouped together into three subsections (pain, emotional states, and general state of health) for the entry version and into four subsections (pain, emotional states, general state of health, and somatic parameters) for the progress version. The pain subsection started with a filtering question, asking whether the patients are currently experiencing any pain. If the patient’s response was negative, they were redirected to the emotional state subsection. However, if their response was positive (i.e., they are currently in pain), they were asked for the location, intensity, duration, and frequency of pain and asked whether this pain is related to the upcoming surgery. In the emotional state subsection, the patients were asked how sad, anxious, tired, numb/dizzy, weak, or irritated they are and how their overall mood is. In the general state of health subsection, the patients stated their general health using the provided answer categories. In the progress version, the patients were only asked about the intensity of pain and whether this pain is related to the surgery. The items of the emotional state and general state of health in the progress version were identical to those in the entry version. The progress version included items pertaining to somatic parameters, in which the patients were asked about their diet and mobility and whether they could have moved/dieted more or less (Appendix A).

### 2.2. Validation

To validate the PHI, it was tested at the Schoen Clinic Hamburg Eilbek, a German hospital clinic.

#### 2.2.1. Participants

Subjects were considered eligible for participation if they were 18 years of age or older and if they had undergone total knee replacement (TKR) as a result of knee osteoarthritis. Those who were cognitively impaired, had insufficient command of the German language, suffered from any mental disorders (according to ICD-10) which require psychiatric medication, consumed any consciousness-impairing substances (e.g., psychoactive drugs, including illegal ones), or suffered from any pain requiring special causative medical treatment (e.g., cancer-related pain) were excluded. However, due to the circumstance that depression and pain are co-prevalent and the population should not be unnecessarily limited, patients with the diagnosis F45.41 (chronic pain disorder with somatic and psychological factors) were not excluded. All patients participated voluntarily and provided written consent.

#### 2.2.2. Study Design and Materials

For recruitment, the patients were first screened for eligibility and then received comprehensive information about the study objectives. Those who agreed to participate and provided written consent were asked to fill in a battery of questionnaires. The first part (entry version) of the PHI was handed to the participants one day prior to the surgery, and the second part (progress version) was handed to them five days after the surgery. All the participants were asked to report their age, gender, marital status, and employment status. For the purpose of validation, von Korff et al.’s questionnaire for grading the severity of pain [58] was assessed. Furthermore, the questionnaire for the assessment of pain-related obstructive self-instructions (FSS) [59], which includes items concerning catastrophizing thoughts (i.e., “I can’t stand the pain anymore” or “When will it be worse again?”) which are rated from (0 = I almost never think this to 5 = I almost always think this), was assessed. The FSS consists of the subscale, negative self-instructions and relates to thoughts and self-talk concerning pain. In addition, the Patient Health Questionnaire-4 (PHQ-4) [60] and an external evaluation by the hospital staff were included. The whole questionnaire battery took approximately ten minutes in total and two minutes for the PHI to be completed. After patients completed the postoperative version, they were asked for comments concerning changes in the inventory. All patients received the same postoperative treatment as usual, which consisted of medication based on pre-existing conditions, physiotherapy, and daily medical visits.

#### 2.2.3. Statistical Analyses

First, representativeness regarding demographic characteristics of the sample was determined. Second, acceptance was assessed by evaluating the answering behavior and examining the missing values. Third, the distribution of values was analyzed using descriptive statistics. Next, sensitivity to changes was analyzed using univariate analyses of variance (ANOVAs) for continuous variables and chi-square test for categorical variables. Validity was analyzed using Pearson’s correlation, and the dimension of emotional state was evaluated using principal component analysis (PCA). For the PCA, Varimax rotation and Kaiser normalization were applied. Missing data were deleted list-wise to avoid data distortions. Extraction and retention of factors were based on a scree plot and an eigenvalue greater than 1.0 [61,62]. The threshold for the Kaiser–Meyer–Olkin (KMO) measure of sampling adequacy was taken as 0.6 [63,64]. To assess the internal reliability of the inventory, Cronbach’s α coefficient values were assessed [65]. To enhance the strength of the factors, only items with a factor loading of at least 0.40 were applied. As an indicator of internal consistency, Cronbach’s α was computed [65]. To assess the items’ sensitivity to changes, ANOVAs were performed. Furthermore, the distribution of scores and the application of each item of the PHI were described.

All analyses were performed using IBM SPSS Statistics version 25.0 (IBM Corp., Armonk, NY, USA). The significance level for all tests was set at α = 0.05 (two-tailed). For correlations, cutoff values were applied to indicate the strength and direction of correlation. Correlations were interpreted using Cohen’s benchmark and categorized as small (|r| = 0.10), medium (|r| = 0.30), or large (|r| = 0.50) [66]. For ANOVAs, partial *η*^2^ was calculated as an indicator of the effect size. The cutoffs of partial *η*^2^ values of 0.01, 0.06, and 0.14 were interpreted as small, medium, and large effects, respectively [66]. The applied cutoff values for Cronbach’s α were graded as follows: acceptable (>0.70), good (>0.80), and excellent (>0.90) [65].

## 3. Results

### 3.1. Patient Characteristics

A total of 135 patients who have undergone TKR at the Center for Orthopedics and Trauma Surgery in Schoen Clinic Hamburg Eilbek, Germany, from October 2015 to March 2019, agreed to participate in this study. Among these, 132 filled in at least one version of the inventory and were, therefore, included in the analysis. These participants are considered representative of those who have undergone TKR from among the German population in terms of age, gender, marital status, and employment status (Table 1).

### 3.2. Acceptance of the Inventory

Overall, the results showed that the inventory was widely well accepted. Not many patients made use of the possibility to comment on the inventory, but declared that they felt well represented. Preoperatively, 91% of the patients responded to the item regarding prominent pain. Moreover, all of those who indicated that they are experiencing pain indicated the painful area preoperatively, and 93% provided information about the painful area postoperatively. Overall, 91% of the patients provided information about the painful area, and 90–92% responded to items pertaining to the emotional state. Information regarding the general state of health was provided by 92% of the patients preoperatively and 94% postoperatively. Furthermore, postoperatively, the patients were asked about their somatic parameters, including their diets and mobility, and whether they were satisfied with them. In total, 86% of the patients responded to items pertaining to somatic parameters, and 87% indicated whether they were satisfied with the diet they received or whether they would have tolerated more or less. In addition, 94% of the patients provided information regarding their mobility, and 93% indicated whether they were satisfied with their amount of mobility or whether they could have moved more or less.

### 3.3. Pain

#### 3.3.1. Painful Areas

In total, 72 patients (55%) stated that they had no other pain besides the preoperative knee pain, whereas 22 patients (17%) pointed out more than one painful area preoperatively besides knee pain. Postoperatively, 87 patients (66%) denied additional pain, 25 patients (19%) reported one additional painful area, and 11 patients (8%) reported several additional painful areas. As all the patients indicated that they were experiencing knee pain related to the upcoming surgery, the additional painful areas were further analyzed. In total, 36 patients (32%) experienced pain in one additional area pre- and postoperatively, 17 patients (15%) experienced additional pain preoperatively but not postoperatively, seven patients (6%) experienced no additional pain preoperatively but some pain postoperatively, and 51 patients (46%) experienced no additional pain either pre- or postoperatively.

Patients who reported additional pain besides knee pain preoperatively were more than those who reported the same postoperatively: χ^2^(1) = 36.41, *p* < 0.001, φ = 0.57. Both pre- and postoperatively, the most prominent painful areas besides the knee were the back, shoulders, arms, and legs. In total, 27 patients experienced back pain preoperatively and 15 patients experienced the same pain postoperatively. Moreover, 19 patients experienced shoulder and arm pain preoperatively and 10 patients experienced the same pain postoperatively. Finally, 18 patients experienced leg pain preoperatively and 10 patients experienced the same pain postoperatively.

#### 3.3.2. Pain Intensity, Frequency, and Duration

The median of the average preoperative knee pain score was found to be NRS = 4 (interquartile range = *IQR* = 3–5), whereas the maximum pain was NRS = 8 (*IQR* = 8–9). The median average postoperative score was found to be NRS = 1.97 (*IQR* = 2–3). All patients reported experiencing knee pain several times a day. In total, 74% of them stated that they have been experiencing knee pain several times a day for more than a year, 15% have been experiencing knee pain for more than 6 months, 8% have been experiencing knee pain for 3–6 months, and only 1% has been experiencing knee pain for less than 3 months.

### 3.4. Factor Loading Pre- and Postoperatively for Emotional States

The PCA revealed particular factor loadings, and the KMO measure of sampling adequacy was found to be 0.81, representing a relatively favorable factor analysis. Moreover, Bartlett’s test of sphericity was found to be significant (*p* < 0.001), indicating that the correlations between items were sufficiently large to perform a PCA. Only factors with eigenvalues of ≥1.0 were considered [62,67]. Examination of the Kaiser criteria and scree plot resulted in an empirical justification for retaining two factors with eigenvalues exceeding 1.0, accounting for 60% of the total variance. Among the factor solutions, the Varimax-rotated two-factor solution yielded the most interpretable solution, and all items loaded highly on only one of the two factors (Table 2). All the data obtained preoperatively loaded on one factor, and all the data obtained postoperatively loaded on another factor. Hence, the results showed that “emotional states preoperatively” loaded on a different factor than that of “emotional states postoperatively.” To perform a reliability analysis, Cronbach’s α was calculated to assess the internal consistency of the “emotional state pre- and postoperatively” subscale, which consists of 14 items. The internal consistency of the inventory was found to be excellent, with a Cronbach α of 0.90 for “emotional state pre- and postoperatively.”

### 3.5. Sensitivity to Changes

The sensitivity to changes was calculated for pain, emotional states and general state of health (Table 3). The pain ratings differed significantly pre- and postoperatively with great effect sizes. For the emotional states, the results showed that the emotional states “sad,” “anxious,” and “irritated” as well as the “general mood” differed significantly pre- and postoperatively with large effect sizes. The patients were found to be significantly more sad, anxious, irritated, and generally less content preoperatively. However, the emotional states “tired” and “weak” did not differ significantly pre- and postoperatively. Finally, general state of health differs significantly. Preoperatively, the patients indicated that their mood was rather good to moderate, whereas after the surgery, they reported an improved state of health that they referred to as good.

### 3.6. Internal Validity

#### 3.6.1. Pain

The item pertaining to pain was found to correlate significantly with that in von Korff et al.’s questionnaire (*r* = 0.34, *p* < 0.001).

#### 3.6.2. Emotional State

The emotional state was found to correlate significantly with pain-related obstructive self-instructions derived from FSS (see Study Design and Materials). In particular, obstructive self-instructions (e.g., “I can’t stand the pain anymore”) were found to exhibit moderately to strongly significant correlations preoperatively with feelings of sadness (*r* = 0.48, *p* < 0.001), anxiety (*r* = 0.37, *p* < 0.001), tiredness (*r* = 0.43, *p* < 0.001), numbness/dizziness (*r* = 0.43, *p* < 0.001), weakness (*r* = 0.53, *p* < 0.001), and irritation (*r* = 0.31, *p* < 0.001), as well as the general mood (*r* = 0.28, *p* < 0.001). Similarly, obstructive self-instructions were found to exhibit moderately to strongly significant correlations postoperatively with feelings of sadness (*r* = 0.41, *p* < 0.001), anxiety (*r* = 0.46, *p* < 0.001), tiredness (*r* = 0.33, *p* = 0.001), numbness/dizziness (*r* = 0.27, *p* = 0.005), weakness (*r* = 0.45, *p* < 0.001), and irritation (*r* = 0.48, *p* < 0.001). Moreover, “sad” was found to correlate with depression from PHQ-4 preoperatively (*r* = 0.58, *p* < 0.001) and postoperatively (*r* = 0.55, *p* < 0.001), whereas “anxious” was found to correlate with general anxiety from PHQ-4 preoperatively (*r* = 0.51, *p* < 0.001) and postoperatively (*r* = 0.63, *p* < 0.001).

#### 3.6.3. General State

The general state of health was found to correlate significantly with the observer rating (*r* = 0.32, *p* < 0.001) of health, and the form of exercise was found to correlate significantly with the observer rating (*r* = 0.22, *p* = 0.02) of exercise.

#### 3.6.4. Somatic Parameters

In total, 73% of the participants considered the diet that they consumed to be a light to normal diet. A total of 1% stated that they received no food, 5% reported receiving a limited amount of water or tea, 7% reported receiving an unlimited amount of tea, 7% reported receiving soup or yoghurt, 1% reported receiving mashed food, and 7% did not provide an answer. Moreover, 18% of the patients stated that they would have tolerated more or other food, 65% were content with their diet, 5% stated that they did not tolerate the food provided, and 11% did not reply to the question.

In total, 73% of the patients were able to move freely without any help. A total of 1% stated that the only form of physical exercise was physiotherapy at the bedside, 3% reported requiring help to walk to the bathroom, 4% reported being able to walk to the bathroom without help, 8% reported being able to move with the help of the hospital staff, 5% reported being able to move with help from other people, and 8% did not reply to the question. Moreover, 17% reported being confident that they would have tolerated more exercise, 72% were content with their level of exercise, and 3% reported being overstrained with their level of exercise.

## 4. Discussion

In this article, we described the development and validation of the PHI, a tool that helps assess the quality of recovery in the perioperative setting. It is highly relevant to increase the quality of recovery in the perioperative setting because the management of postoperative pain is still inadequate [20]. To assess the quality of recovery, this developed tool focuses on the quality criteria that are considered the most important for the patients: pain, general state of health (including emotional states), and somatic parameters (mobility, nourishment). Moreover, it covers the whole perioperative setting by focusing on patient-relevant criteria and also enables a regular comparison. It also assesses the outcome measures of the patients themselves and provides a method for external assessment so that the outcome measures can be evaluated from different perspectives.

When applied regularly, this tool could potentially assist in directing adequate postoperative support to manage pain. This is because to improve postoperative pain management, it is necessary to overcome the causes of inadequate management of acute pain [68]. One cause is that because pain is a multidimensional phenomenon [69,70], to manage postoperative pain effectively an interdisciplinary (e.g., pain therapists, surgeons, pain psychologists) [71] approach is important. However, the organization and the exploration of necessary support can be challenging. With this tool, the treating team can more easily assess which profession is needed and request further and necessary support to manage postoperative pain effectively. Hence, the PHI could overcome this obstacle because it includes different dimensions of pain management. For example, the pain scales would especially be relevant for pain therapists, the emotional states for psychologists, the mobility question for physiotherapists, and the nutrition items for dietitians. Thus, in general, the PHI is considered a valuable tool because adequate assessments and evaluations are the only means for an interdisciplinary postoperative treatment that not only has a direct effect on the immediate treatment outcome but also decreases the possibility of negative long-term effects.

The frequency of the application of the PHI depends on clinical need and purpose. In general, daily or near-daily use would be helpful to assess the clinical progression of treatment and healing. Therefore, if it is reasonable, daily or almost daily application of the PHI is recommended. Due to the reason that the inventory can be filled in within two minutes, it might be more time efficient to apply this tool instead of conducting a personal interview containing the same content. The results obtained in this study show that patients who undergo TKR report less pain postoperatively than preoperatively (baseline assessment) and are also more emotionally stable. This is in line with previous research findings, which have indicated that an upcoming surgery entails psychological stress for patients preoperatively [72,73] but that this stress decreases postoperatively [74,75]. In this study, we were able to focus on emotional states and to include different aspects pertaining to the emotional state without requiring a large number of questionnaires. It is also possible to evaluate the emotional states of patients who have a tendency to have depression or anxiety but are not psychopathologically depressed or do not have an anxiety disorder.

### 4.1. Strengths of the Study

Besides its primary aim of developing an inventory for assessing patient-relevant outcomes to evaluate the quality of recovery, the present study provides an inventory that helps assess whether an additional intervention related to pain treatment, emotional support, or changes in mobilization or diet is necessary and should be introduced. This is considered highly relevant because of the known lack of adequate postoperative pain treatment [20]. The different subscales revealed favorable psychometric properties in terms of factor loadings, excellent Cronbach’s α values, and other relevant measures. In general, it was found that the PHI is a potentially useful tool that can be applied in the perioperative setting to compare the quality of recovery between different health care providers and different types of surgery. Within the development phase, the patients’ perspective was thoroughly included; in each development step, patients were encouraged to state their opinion and their statements were then included in the modification of the PHI. Hence, it can be concluded that this inventory truly includes aspects that are considered the most relevant for the patient within the perioperative setting. This inventory was tested to ensure that it is easy to understand by all patients, reflects the patients’ personal perspectives, is well accepted, maps changes in the perioperative period, and is easy to apply in everyday clinical practice. Generally, the PHI is considered the first inventory that is particularly designed for the perioperative setting, and it allows the assessment of the quality of recovery and helps point out whether further immediate treatment is required. This may help prevent negative long-term consequences, such as chronic pain or decreased quality of life, which could have positive economic consequences.

### 4.2. Limitations of the Study

Given the chosen target group, the average age in this study was relatively high (68 years). The age is a limiting factor because the results cannot without further studies be generalized to all age groups. However, for the investigated patient group, the age is representative. Moreover, for the purpose of the final validation, only patients who underwent TKR were considered, whereas for previous validation general surgery patients were considered. Furthermore, anesthesia-relevant effects are not included in the inventory, but might also be relevant for further treatment. One limitation of the inventory is that only the average pain is asked, but is not differentiated between pain at rest and pain at movement. Therefore, for a more detailed assessment, in the most recent edition of the inventory items concerning pain at rest and pain at movement were added.

### 4.3. Future Research Steps

To generalize the results to other types of surgery, the PHI should be implemented for different types of surgery and age groups and the results should be confirmed using confirmatory factor analysis. Including the results obtained with other types of surgery can help establish and accurately evaluate the differences between the types of surgery and quality standards. Another planned research step is to merge the PHI with a questionnaire which focuses primarily on negative clinical effects of the anesthesia (e.g., nausea, dyspnea). A study is currently ongoing on this instrument (Fischer, M., Zöllner, C. https://clinicaltrials.gov/ct2/show/NCT04528537?term=quality+of+recovery&cond=quality+of+recovery&draw=2&rank=3, accessed on 29 March 2021).

## 5. Conclusions

The PHI is a general tool that focuses on a detailed evaluation of pain and the state of health within the perioperative setting. It allows the assessment of the quality of recovery and helps guide immediate adequate interdisciplinary postoperative pain treatment. Therefore, it is recommended to apply the PHI in the perioperative setting because it is easy to understand, is well accepted, maps changes in the perioperative setting, and includes patient-relevant parameters.

## Figures and Tables

**Figure 1 jcm-10-01965-f001:**
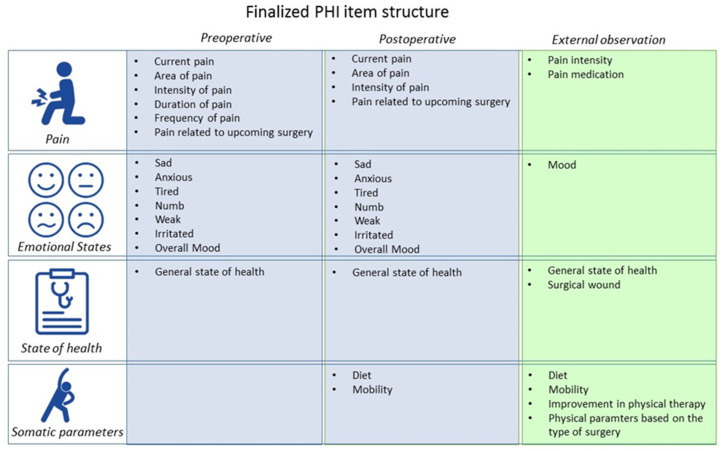
The item structure of the pain and state of health inventory (PHI).

**Table 1 jcm-10-01965-t001:** Demographic characteristics and baseline characteristics related to age, gender, marital status, and employment status.

Item	Sample (*n* = 132)
Mean age, years (SD)	68 (9.4)
Male/female	58/74
Marital status	
Single	10
Married	88
Widowed	18
Divorced	11
Serious relationship	3
Living separately	1
Missing	1
Employment	
Incapacitated to work	7
Unemployed	4
Employed	32
Retired due to illness	10
Retired due to age	75

**Table 2 jcm-10-01965-t002:** Varimax-rotated component matrix for the factor loading of emotional states.

Item	Component	
	1	2
Postoperatively		
General mood	0.862	
Sad	0.798	
Anxious	0.779	
Weak	0.733	
Irritated	0.720	
Numb/dizzy	0.684	
Tired	0.672	
Preoperatively		
General mood		0.840
Weak		0.811
Tired		0.758
Numb/dizzy		0.711
Sad		0.697
Anxious		0.664
Irritated		0.654

**Table 3 jcm-10-01965-t003:** Pre- and postoperative indicators for sensitivity to changes in emotional states.

Item	*F*	*p*	Partial *η*^2^	Mean_Pre_ (SD)	Mean_Post_ (SD)
Pain					
Pain	42.33	<0.001	0.28	3.95 (2.58)	2.09 (1.91)
Emotional state					
Sad	29.03	<0.001	0.21	2.00 (2.25)	1.01 (1.47)
Anxious	50.45	<0.001	0.32	2.23 (2.43)	0.71 (1.20)
Tired	1.10	0.30	0.01	2.30 (2.12)	2.16 (2.03)
Weak	0.97	0.33	0.01	1.95 (2.24)	1.74 (1.94)
Irritated	21.18	<0.001	0.17	1.50 (1.85)	0.67 (1.20)
General mood	42.17	<0.001	0.17	2.05 (2.00)	1.16 (1.51)
State of health					
State of health	20.22	<0.001	0.15	2.11 (1.04)	1.61 (0.94)

## Data Availability

Data are available on request due to privacy and ethical restrictions.

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
