# Peer review of "Development and Validation of the Pain and State of Health Inventory (PHI): Application for the Perioperative Setting"

_jcm, 2021, doi:10.3390/jcm10091965_

Round 1

Reviewer 1 Report

Thank you for offering your submission for review.

This is an interesting topic and its good to see a paper exploring better post operative pain management.

Perhaps your title or at least the first sentence in the abstract (or both) should say that the PHI is for post op pain?

Abstract Ln19 maps 

Ln21 "allow" might be a better word than achieve

Introduction

You make a good and sensible case for exploring other (?better) post op pain assessment tools.

Ln 47 "legal" is a very strong word to apply to a guideline (ie Guidelines aren't mandatory!) I suggest remove

Ln 59 in English a better word is practice ((for praxis) you use this word later as well).

Ln 87 I see your group developed this tool initially in 1999 (ref33) have you used it regularly since then or is this work you are reporting now a rediscovery of this previous work?

Materials and Methods

Ln 106 you use the word expert several times in your development process. Who were these experts and how were they defined.?

Im not sure Fig 1 helps much

Somatic parameters

You discuss diet with two references. Clearly the metabolic state of patients is important but in your results and the PHI as shown in the Apdx might be measuring palatability of food and be a reflection on the hospital's kitchen rather than nutrition in the early post op period after TKR. In an uncomplicated TKR do you think that nutrition in very important?

General comments

Are you planning to offer this post op questionaire daily?

Clearly the results will be different as pain and mobility improve. Will your PHI be sensitive enough to pick up these changes

Along with diet and nutrition did you consider nausea?

Table 2

I am not familiar with a Varimax matrix and the Table did not present properly on my pdf but I noted that preop anxious was .664 and post op anxious was .779. Are higher scores better? In the text you talk of anxiety scores reducing (as expected)

How long does the questionaire take to complete?

Did your clinicians especially the ward nurses assessing post op pain like it?

In the hospital I work in the Acute Pain Nurses do pain scores at every shift change and after administering analgesia. Will  you recommend this tool be used very frequently or do you have a second NRS system?  

Reviewer 2 Report

The authors are to be commended on their efforts to develop a tool for assessing pain, health, and quality of recovery in the perioperative setting that is consistent with a biopsychosocial approach. While there are several available measures in circulation that address specific aspects of these constructs, the motivation here was to develop a single tool of comparable psychometric status that can be simply administered in the pre- and post-operative setting to drive immediate treatment response. The gathering of patient and provider perspectives in attempting to validate the PHI was a particular strength of the methodology.

The current study and manuscript do have significant limitations that need to be addressed:

  • In navigating through the appropriate steps of measure development, the authors cite changes made to the PHI instrument following steps 1 and 2, yet there are no details provided about those changes. This information should be described for the reader. Similarly, the authors cite a “final validation” step on page 5 of the manuscript, yet no details are provided. Is this an advanced reference for what comes later in the manuscript?
  • Further details about the members of the “expert panel” the authors used in the initial development of the PHI should be detailed for the reader, including the clinical disciplines represented by those experts. This panel should represent multiple clinical perspectives that are being tapped by the PHI.
  • In the validation phase of the instrument, the authors excluded individuals who, “suffered from any mental disorders.” The rationale for this exclusion was not made clear, and in light of the significant co-prevalence of pain and depression/anxiety, the exclusion of such individuals unnecessarily limits the population to whom the PHI applies. It is also unclear if this same exclusion criteria was utilized in the first two samples used to guide measure development.
  • The authors used participant completion of the PHI as the measure of “acceptance” for the instrument. It can be argued that completion of a measure does not reflect acceptance of the same, particularly in a research context. Acceptance would be better assessed via direct questioning of the participants about their experience with the measure. In the preliminary study the authors cited a rating process whereby the participants rated the PHI and the questionnaires used for validation in a direct way. This rating process and the rating results should be more thoroughly described, and a similar process should have been utilized in the assessment of acceptability of the final version of the PHI.
  • In describing the internal validity results for the emotional state section of the PHI, the authors cite correlation with, “obstructive self-instructions”, yet no explanation of obstructive self-instructions was provided. More details should be provided here.
  • The sensitivity to change section of the manuscript was not written very clearly, and appears to mix some of the findings between emotional states and state of health. This section should be re-written to make the case for sensitivity to change more clear to the reader.
  • The aim of the study included assessing if the PHI was, “economically suitable” for use in clinical practice. However, there were no such assessments included in the manuscript.
  • In the conclusion, the authors cite how the PHI, "helps guide immediate adequate interdisciplinary postoperative pain treatment," yet this was not directly tested as part of this study. This conclusion should be further explained. Is a formal instrument necessary to get at some of these constructs in order to direct immediate changes in care?

Round 2

Reviewer 1 Report

Thanks for resubmitting and adressing my comments.

Minor suggestions now

Ln 328 bur should be but

Ln242    F45.41 it would be helpful to add the text that this code refers to
